# Psychopathology, Protective Factors, and COVID-19 among Adolescents: A Structural Equation Model

**DOI:** 10.3390/ijerph20032493

**Published:** 2023-01-30

**Authors:** Christin Scheiner, Christian Seis, Nikolaus Kleindienst, Arne Buerger

**Affiliations:** 1Department of Child and Adolescent Psychiatry, Psychosomatics and Psychotherapy, University Hospital of Würzburg, 97080 Würzburg, Germany; 2German Center of Prevention Research in Mental Health, 97080 Wuerzburg, Germany; 3Department of Psychology I, Wuerzburg University, 97070 Würzburg, Germany; 4Central Institute of Mental Health, Department of Psychosomatic Medicine and Psychotherapy, Medical Faculty Mannheim, Heidelberg University, 68159 Mannheim, Germany

**Keywords:** adolescence, mental health, psychopathology, protective factors, COVID-19

## Abstract

Since the outbreak of the COVID-19 pandemic in December 2019 and the associated restrictions, mental health in children and adolescents has been increasingly discussed in the media. Negative impacts of the pandemic, including a sharp increase in psychopathology and, consequently, reduced quality of life, appear to have particularly affected children and young people, who may be especially vulnerable to the adverse effects of isolation. Nevertheless, many children and adolescents have managed to cope well with the restrictions, without deterioration of their mental health. The present study therefore explored the links between COVID-19 infection (in oneself or a family member, as well as the death of a family member due to the virus), protective factors such as self-efficacy, resilience, self-esteem, and health-related quality of life, and measures of psychopathology such as depression scores, internalizing/externalizing problems, emotion dysregulation, and victimization. For this purpose, we examined data from 2129 adolescents (mean age = 12.31, SD = 0.67; 51% male; 6% born outside of Germany) using a structural equation model. We found medium to high loadings of the manifest variables with the latent variables (COVID-19, protective factors, and psychopathology). Protective factors showed a significant negative correlation with psychopathology. However, COVID-19 had a weak connection with psychopathology in our sample. External pandemic-related factors (e.g., restrictions) and their interaction with existing psychopathology or individual protective factors appear to have a greater influence on young people’s mental health than the impact of the virus per se. Sociopolitical efforts should be undertaken to foster prevention and promote individual resilience, especially in adolescence.

## 1. Introduction

In March 2020, the World Health Organization (WHO) [1,2] declared the severe acute respiratory syndrome coronavirus 2 (SARS-CoV-2) as a pandemic. First detected in December 2019, the virus spread rapidly [3], reaching over 500 million infections and around six million deaths worldwide by July 2022 [4]. Great hope was placed in vaccination, which was available in Germany from December 2020 [5]. The chronology of the pandemic in Germany, including the dominant types of the virus, infection rates, and vaccine availability is illustrated in Figure 1.

In response to the outbreak, countries around the world imposed protective measures such as lockdowns, quarantines, school and workplace closures, social distancing, contact restrictions, and the wearing of facemasks in public. However, these restrictions to everyday life came with a variety of negative consequences [6,7,8], with current research showing a significant and sharp deterioration of mental health [7,8,9,10]. Young people seem to be especially vulnerable to the adverse effects of isolation, as they show a higher risk of developing mental health problems [11]. A recent meta-analysis examining the impact of COVID-19 on mental health in children and adolescents (mean age  =  11.3 years, 49.7% female) found a prevalence of anxiety and depressive symptoms ranging from 1.8–49.5% and 2.2–63.8%, respectively [12]. Longitudinal research has demonstrated a rise in anxiety and depressive symptoms compared to before the lockdowns, with longer lockdowns having a significantly greater impact on mental health in at-risk groups [12]. Another recent study found that exposure to pandemic-related stressors was associated with higher levels of internalizing problems among adolescents during the pandemic [13]. Studies investigating internalizing and externalizing problems have reported that social connectedness represents a protective factor against increased psychopathology [14,15]. Therefore, we sought to take a closer look at depression scores and internalizing/externalizing problems in young people during the pandemic.

Since the start of the pandemic, research has been examining risk and protective factors in adults and adolescents. In a large sample of adults in China, no significant changes in depression or anxiety over time were found [16]. With regard to adolescents, research has reported increased concerns about COVID-19, loneliness, feeling disconnected from friends, and social media use. On the other hand, sport/exercise was found to mitigate the negative effects of the pandemic among adolescent males, as did spending more time with family and friends [17,18,19,20]. Compliance with stay-at-home orders and social connectedness during the COVID-19 lockdown have been identified as protective factors for mental health. Early longitudinal evidence suggests that adolescents are more concerned about government restrictions than about the virus itself, and that these concerns are associated with a deterioration of their mental health [20]. Moreover, a Spanish study reported that adolescents who had been infected with COVID-19 were more likely to remain mentally healthy [21]. In the present study, we therefore examined to what extent only infection-specific factors, such as one’s own infection, or infection or death in the family, have impacted adolescents’ mental health.

Although children and adolescents were generally at a lower risk of infection and less likely to become severely ill with COVID-19 [22], they have nevertheless been tremendously affected by the pandemic, mostly due to the social restrictions [23,24]. Rider et al. (2021) [25] summarized stress factors that rendered the pandemic particularly challenging for children and adolescents’ mental health and well-being (see Table 1). It should be noted that these stress factors do not usually act and occur alone, but rather interact, and thus become amplified, with relevant implications for, e.g., emotion regulation [26,27]. Adaptive emotion regulation with adaptive coping is associated with lower levels of psychopathology. Accordingly, maladaptive coping, such as suppression, avoidance, and denial, has a reinforcing effect on psychopathological symptoms [28]. With regard to COVID-19, studies have demonstrated that emotion dysregulation increases the risk of persistent negative mental health [29] and that reducing pandemic-related stressors and promoting adaptive emotion regulation strategies can exert protective effects on adolescents, especially in times of increased stress [13]. For these reasons, we also included emotion dysregulation in our analysis.

Despite the stress factors (displayed in Table 1), which could have affected all children and adolescents equally, most have managed to get through the pandemic well and have maintained their mental health. Particularly in the case of highly dynamic and stressful events, such as a pandemic, it is essential to better understand the role of protective factors [24,30]. Research has already begun to determine which children and adolescents are at particular risk of deteriorated mental health. Health-related quality of life (HRQoL), for instance, has become a major health outcome examined in epidemiological research in children and adolescents, and reflects physical and psychological well-being, family life, school environment, and peer relations [31]. Girls were found to report a lower HRQoL than boys, especially in adolescence. Furthermore, a low socioeconomic status (SES) and greater mental health problems have been associated with a low HRQoL. In contrast, self-efficacy, self-esteem, and social support are reported to have protective effects on HRQoL [32,33]. For these reasons, we also investigated HRQoL, self-efficacy, self-esteem, and SES in the present study.

In sum, the present study sought to find out more about the effects of protective factors as a consequence of COVID-19 in a sample of adolescents. For this purpose, we used a structural equation model (SEM) [34] with manifest variables based on our dataset and in line with the literature regarding protective factors (self-esteem, self-efficacy, resilience, and HRQoL), the influence of COVID-19 (infection of self and family), and psychopathology (externalizing and internalizing problems, emotion dysregulation, and depression scores). With this model, we wished to gain deeper insights into the complex interplay of these different constructs in order to inform further and targeted research or prevention measures and support this group of children and adolescents.

## 2. Materials and Methods

### 2.1. Participants & Procedure

In fall 2021 and spring 2022, 2129 adolescents aged 11–14 years (mean = 12.31, SD = 0.67) were recruited from 18 different high schools in Germany. The utilized data stem from the baseline assessment of a longitudinal RCT called DUDE (“You and Your Emotions”, German: “Du und deine Emotionen”). Data collection took place within the aforementioned timeframes and data were not assessed continuously. The participants completed all questionnaires in the classroom under supervision by the study staff, using their own mobile phones or school-owned tablets to access the survey platform REDCap [35]. Inclusion criteria were being in the 6th or 7th grade and the provision of informed consent from parents/legal guardians and the adolescents themselves [36]. The study was approved by the responsible Ethics Committee of the University of Wuerzburg.

### 2.2. Questionnaires

#### 2.2.1. To Measure the First Latent Variable “Protective Factors”

HRQoL: The KIDSCREEN-10 Index was used to generate the index for Health-Related Quality of Life. It was derived from the 27-item version and operationalizes general HRQoL in the last week. The index has shown good internal consistency (Cronbach’s α = 0.82) and good test-retest reliability (r = 0.73; ICC = 0.72) [37,38].

Self-efficacy (S_p/S_d/S_a): The revised Regulatory Emotional Self-Efficacy Scale (RESE-R) was used to assess self-efficacy in general, without a given timeframe [39,40]. It contains 12 items rated on a 5-point Likert scale from 1 (=not at all good) to 5 (=very good) and is composed of two dimensions: one for perceived self-efficacy in expressing positive emotions (POS) and the other in expressing negative emotions (NEG). The negative dimension is further subsumed into self-efficacy in dealing with dejection/despair (DES) and self-efficacy in dealing with anger/irritation (ANG). Good internal consistencies have been reported for the individual domains (α = 0.68 to 0.79) [39].

Self-esteem (SE): To assess self-esteem, we used the Single Item Self-Esteem scale (SISE; [41]). The SISE includes the item “I have high self-esteem”, which is rated on a 5-point Likert scale from 0 (=does not apply at all) to 4 (=applies very much) referring to the last week. The reliability lies at 0.75 [41].

Resilience (Re): The Connor-Davidson Resilience Scale 10 (CD-RS 10, [42]) was used to measure resilience in general, without a given timeframe. It comprises of ten items rated from 0 (=not at all true) to 4 (=almost always true). The items deal with resilience to stressful events such as illness, painful feelings, and stress. A high total score is interpreted as a high ability to recover well from stress.

#### 2.2.2. To Measure the Second Latent Variable “Psychopathology”

Internalizing and externalizing problems (int./ext.): Two subscales of the Strengths and Difficulties Questionnaire (SDQ) were used to assess internalizing problems (emotional difficulties and peer problems) and externalizing problems (behavioural problems and hyperactivity) during the last six months [43,44]. The SDQ is a widely used diagnostic instrument for measuring behavioural problems and behavioural strengths in children and adolescents aged from 3 to 16 years using 25 items (items are rated as “not true”, “somewhat true”, or “certainly true”) [45].

Depression (De): To assess depressive symptoms during the last two weeks, we used the nine-item Patient Health Questionnaire (PHQ-9) [46], which is a screening instrument based on the Diagnostic and Statistical Manual of Mental Disorders, Fourth Edition (DSM-IV [47]. The items are rated on a 4-point Likert scale, with scores higher than 10 indicating probable depression (range = 0–36). The measure is widely used among adults and has been validated for adolescents, showing good psychometric properties (sensitivity of 89.5% and specificity of 77.5% for detecting adolescents with major depression; Cronbach’s α = 0.83, rtt = 0.87) [48].

Emotion regulation difficulties (Emt): The DERS-SF [49], an 18-item short version of the original 36-item Difficulties in Emotion Regulation Scale [50,51], was used to assess difficulties in emotion regulation in general without a given timeframe. The DERS-SF shows good psychometric properties, with a Cronbach’s alpha of 0.98 for the total score [52].

#### 2.2.3. To Measure the Third Latent Variable “Impact of COVID-19”

Covid-infection (fml/pos): To assess the impact of COVID-19 on the participants, two questions were derived from the Robert Koch Institute (RKI) Corona-Warn-App to measure individual exposure to the pandemic within the last year. Questions were extracted based on how well they fitted our aim of exploring the pandemic’s influence on the adolescents within their families [53]. The questions were “Have you tested positive for the COVID-19 virus?”, and “Has someone in your family tested positive for COVID-19?”.

### 2.3. Statistical Analysis

As psychology contains many constructs that are subject to measurement error, such as stress or resilience, the analysis of latent variables is recommended [54]. A structural equation model [34] consists of a measurement model with exogenous/manifest variables and a structural model with endogenous/latent variables [55]. To test our hypotheses, we ran a series of structural equation models (SEMs) in R [56,57] using the lavaan package [58]. All analyses were conducted using latent variables for the impact of COVID-19, protective factors, and psychopathology, following similar approaches to those used elsewhere [59]. A key advantage of a latent SEM approach is that it enables one to model measurement error [60].

After checking for normality, multicollinearity, and outliers, and transforming the data (one variable was not linear; we also standardized the whole data frame for easier comparison), we calculated the first SEM model with minimal structure. Cases with missing data were excluded from the analysis (total of missings = 263, distributed across 14 variables). In summary, we constructed two different SEM models. The first contained only the measurement model consisting of three equations and one regression. To the second model, we added three residual correlations to improve model fit. Models were estimated using the maximum likelihood method, which is a rather robust method in the case of outliers (MLM; [61]). To minimize the impact of outliers, we further bootstrapped the final model to obtain confidence intervals. Model estimates were also standardized. *p*-values smaller than 0.05 were considered statistically significant.

### 2.4. Research Question and Hypotheses

As outlined above, we examined the impact of COVID-19, protective factors, and adolescents’ psychopathology. The hypothesized model is presented in Figure 2. We hypothesized a positive correlation of COVID-19 infection on psychopathology (the more virus, the more psychopathology) and a negative correlation of protective factors on both psychopathology and on the impact of Covid-infection (the more protective factors, the less psychopathology or virus related impact). The manifest variables (see Section 2.2) measured latent variables such as the impact of COVID-19, protective factors (HRQoL, self-esteem, self-efficacy, and resilience), and psychopathology (externalizing and internalizing problems, emotion regulation difficulties, and depression score). With this model, we wished to contrast the complex relationships between the external adversities of COVID-19 and individual internal resilience such as protective factors and measures of psychopathology.

### 2.5. Measures of Model Fit

In general, a chi-square (χ^2^) test is used to assess model quality. This test examines the deviation of the sample variance-covariance matrix from the variance-covariance matrix of the estimated model in inferential statistics. A non-significant deviation indicates acceptable model quality, i.e., a model that fits the data well. However, the chi-square (χ^2^) deviation test is very sensitive to the sample size. Since this was the case in our sample (more than 400 cases), we relied on other indices. Specifically, we used the Tucker-Lewis index (TLI) and the comparative fit index (CFI) as incremental measures of quality. These indices compare a null model with the postulated model. Model fit is deemed to be acceptable if the TLI is greater than 0.9, and good if the TLI is greater than 0.95 and the CFI is greater than 0.95 [62,63]. As an absolute index, we used the root mean square error of approximation (RMSEA). A common threshold value for the RMSEA reported in the literature is 0.06, which indicates a good model fit to the data. A value of 0.08 is considered average/acceptable [62]. The standardized root mean square residual (SRMR) is another absolute measure of quality, with a value of less than or equal to 0.08 denoting a good model fit [62,64]. Furthermore, we used the Akaike Information Criterion (AIC) and the Bayesian Information Criterion (BIC), with smaller AIC or BIC values suggesting a better model fit. For the improvement of model fit, we included covariances in the next steps.

## 3. Results

### 3.1. Sample Description

The sample was comprised of 2154 adolescents with a mean age of 12.31 years (SD = 0.67). A total of 51.1% were 6th grade students, 51% were male, and 12 adolescents stated that they were gender diverse. One tenth of the sample was not born in Germany. See Table 2 for mean scores and standard deviations.

### 3.2. Correlation Matrix

A correlation matrix of the variables included in the SEM is shown in Table 3. The factor intercorrelations indicate positive relationships among the protective factors themselves (HRQoL, self-efficacy, self-esteem, and resilience) and negative relationships between psychopathological factors (externalizing and internalizing problems, depression score, emotion dysregulation, and bullying/victimization) and protective factors. The relationships among the COVID-19 measures indicate the strongest correlation between infection of oneself and infection of a family member. The correlations of COVID-19 measures with protective factors and psychopathological measures were small.

### 3.3. SEM

#### 3.3.1. Final Model

The first model tested (model 1) indicated an acceptable fit to the data according to all criteria: χ2 (51, N = 2129) = 825.034, *p* < 0.001, CFI = 0.885, TLI = 0.851, RMSEA = 0.084, SRMR = 0.060, AIC = 61,935.813, and BIC = 62,088.725 (note: for all indices, we used the robust parameters/the adjusted parameters). An alternative model 2 was tested including covariances for the three subscales of self-efficacy, which indicated a slightly better fit to the data: χ2 (48, N = 2129) = 645.790, CFI = 0.919, TLI = 0.889, RMSEA = 0.097, SRMR = 0.050, AIC = 61,519.747, and BIC = 61,689.649. A scaled chi-square difference test (using the method “Satorra Bentler”) showed a significant superiority of model 2 (χ^2^(3) = 97.35, *p* < 0.001). Given the large number of outliers in the data frame and the fact that not all variables were perfectly normally distributed, we additionally bootstrapped model 2, leading to the following final result: χ^2^ (48, N = 2129) = 1045.697, *p* < 0.001, CFI = 0.917, TLI = 0.886, RMSEA = 0.099, and SRMR = 0.050. All hypothesized paths were significant and in the expected direction, except for the ones between Covid-infection positive and Covid-infection family as protective factors. Our final SEM is visualized in Figure 3.

#### 3.3.2. Overall Effects

A significant negative association was found between protective factors and psychopathology (A: β = −0.57, *p* < 0.001; r = −0.80) and a significant positive association emerged between impact of COVID-19 and psychopathology (B: β = 0.10, *p* < 0.001; r = 0.08). Protective factors and COVID-19 were not significantly associated (C: β = 0.04, *p* = 0.68, r = 0.01).

#### 3.3.3. Loadings on Manifest Variables

In terms of the loadings on adolescent psychopathology, this latent variable showed moderate to strong significant correlations with externalizing problems (r = 0.57) and internalizing problems (r = 0.79), emotion dysregulation (r = 0.68), and depressive symptoms (r = 0.87).

The factors self-esteem (r = 0.80), resilience (r = 0.76), self-efficacy (r = 0.62/0.71/0.59), and HRQoL (r = 0.80) also showed strong and significant correlations with the latent variable protective factors.

Finally, the latent variable Covid-infection showed significant correlations with the manifest variables own infection (r = 0.46) and with the infection of a family member (r = 0.92). 

## 4. Discussion

The present study examined the relationship between protective factors and psychopathology as influenced by COVID-19 (infection of oneself or a family member, death of a family member). Using a structural equation model, we found a strong negative correlation between the protective factors and psychopathology in our sample, suggesting that adolescents who scored high on, for instance, resilience or self-efficacy were in good mental health. Despite good loadings of the manifest variables measuring the latent variable impact of COVID-19, it only showed a very small positive correlation with the latent variable psychopathology. This finding is in line with recent research suggesting that adolescents are more concerned about government-mandated restrictions than about the virus itself. Furthermore, this study found adolescents more likely to remain mentally healthy when they are infected with the virus [20,21]. We cannot confirm this finding with our data; however, only a minority of this sample had a COVID-19 infection. Moreover, the correlation between the protective factors and the impact of COVID-19 was zero, suggesting no mutual influence of these two latent variables. This finding does not confirm our suspected influence between protective factors and COVID-infection, but rather suggests that being infected with the virus is not associated with resilience, self-efficacy, or quality of life. Hence, the observed deterioration in mental health that has started with the outbreak of COVID-19 and the following restrictions might rather be attributed to, e.g., the stress factors summarized and suggested by Rider et al. (see Table 1) [25]. In sum, the restrictions due to the virus, and not the virus itself or virus-related fears, seem to be responsible for the deterioration in adolescents’ mental health over the last two years. The pandemic has led not only to social isolation but also to disrupted routines (altered school schedule, sleep patterns, and fewer leisure activities). Children and adolescents have undertaken less physical activity, their media use has frequently increased, and in some cases, they have even witnessed or suffered increased domestic conflicts and violence (perhaps even trauma), resulting in a lower ability to regulate stress and emotions [23,65,66]. However, there is reason to believe that the elevated levels of psychopathology (depression, internalizing and externalizing problems, and emotion regulation difficulties) in the adolescents in our sample are unlikely to be explained by an infection with COVID-19 (self, family, or death in the family). Therefore, it is particularly important to investigate the impact of COVID-19 restrictions on mental health.

In terms of the manifest variable psychopathology and its correlations, we found medium to high correlations between externalizing problems, internalizing problems, depression, emotion dysregulation, and psychopathology. This is in line with recent research suggesting that an increasing number of children and adolescents have been suffering from psychological symptoms since the beginning of the COVID-19 pandemic [67,68]. These meta-analyses mention closed schools and social distancing, among other things, as the main stress factors for families during the pandemic, which have in turn also caused psychological problems in the caregivers [67,68]. The findings are partially consistent with our hypothesis, that the reasons for a deterioration in mental health are not directly related to COVID-19 infections per se and must be sought elsewhere. As described in the Introduction, a number of social restrictions were introduced in Germany during the pandemic, including lockdowns and school closures. School closures were—and remain—a matter of debate in research and politics, as the relative degree of effectiveness and efficiency of this measure, when balanced against the adverse effects on students’ mental health and school attainment, remains unclear. A recent study examining school closures and school reopening in Germany during the pandemic [69] reported that especially the youngest of the sample (11–13 years) suffered most in terms of decreased HRQoL and increased psychopathology, which is also in line with findings from Ravens-Sieberer et al. (2021) [24]. In summary, it seems that school routines represent very important coping mechanisms for children and adolescents, specifically those with mental health problems [70], and the long-term effects on these individuals, as well as on young people with no prior mental health issues, should be investigated in more detail in the future.

Looking at our results regarding the protective factor, we see that the correlations between the individual manifest variables are medium-sized (r = 0.45−0.66). The loadings on our latent variable are medium to large (r = 0.59–0.80). From this we conclude that the interaction of individual protective factors could be decisive with regard to salutogenesis and, consequently, future prevention programs. Furthermore, we have to consider that the quality of life, as well as self-efficacy in relation to positive emotions and self-esteem, are quite high in our sample. This could mean our sample was reported to have a rather high quality of life. When looking at the means of external/internal problems, emotion dysregulation, and depression, it seems as if our sample did show very low mental health problems to begin with. Surprisingly, resilience in particular is rather low on average (M = 24.70), because a value of <23 can be considered as clinically relevant [71]. Accordingly, one conclusion to draw for universal prevention is that it might make sense to specifically strengthen the resilience of students. Targets like coping with stress, to deal with failure, to stay focused under pressure, or to handle unpleasant feelings are recommendations for future prevention trainings, especially in a general and mixed population, which you find at schools. This conclusion is in line with a systematic review of resilience-focused interventions that found such interventions to be effective in terms of reducing depressive symptoms, internalizing problems, externalizing problems, and general psychological distress [72]. Evidently, not only does the timing of the intervention (childhood or adolescence) seem to play a role, but also which specific factors one seeks to strengthen and whom one includes (e.g., the parents). This question of when, who, and how is crucial in terms of prevention. The meta-analysis mentioned in the Introduction of paper [12] identified routines, parent-child discussion/communication/relationship, play, and physical activity as protective factors for mental health during the COVID-19 lockdown. According to the literature, younger boys coped the worst with the pandemic situation [24,69]. Therefore, gender differences should be taken into account in schools and in terms of prevention programs/interventions. Indeed, there is some evidence suggesting different coping and mental health among girls and boys in childhood and adolescence [73,74].

### Limitations

Some limitations of the present study should be mentioned. While we focused on protective factors and revealed their relevance in adolescence, we did not analyze factors such as reduced physical activity, limited social experiences, parental stress, family conflict/domestic violence, and so on. Future research should therefore investigate these factors. Although our sample is quite large, with over 2000 adolescents, the age range (11–14 years) and examined region (state of Franconia) are limited. Furthermore, as we collected data at two time points (fall 2021 and spring 2022), our data are likely biased due to differing pandemic-related restrictions at these time points. With regard to our assessment of the impact of COVID-19, our design could have been more elaborate. Rather than using a dichotomous design (infection: yes/no), it would have been useful to ask about thoughts and fears regarding COVID-19, when/how long ago infections occurred, and the severity of illness. This would have enabled us to generate a variable including these aspects and represent interindividual affectedness. In addition, we did not assess any risk factors in our sample, which would have improved this SEM. Considering the assessment of victimization, the reliability of this variable might be decreased in terms of the altered school life at data assessment, plus adolescents were not given a strict definition of bullying/victimization beforehand. Furthermore, we only interviewed students at German grammar schools, which are usually associated with a higher functional level of the students and a higher socioeconomic status. Accordingly, the generalizability of our results is limited. The floor effect in relation to deaths from infection with the COVID-19 virus, as mentioned in the Results section, entails statistical difficulties. Moreover, no causal conclusions can be drawn from our SEM. Longitudinal data that consider changes over the course of the pandemic would be desirable. Longitudinal studies would be appropriate at this point in order to make truly causal statements about an improvement and specific promotion of resilience or self-efficacy.

## 5. Conclusions

In summary, empirical evidence regarding the consequences of the COVID-19 pandemic demonstrates a strong psychological burden in terms of increased pandemic-related stress, lower emotion regulation ability, increased depression and anxiety, and a lack of social skills training in children and adolescents [9,29]. A recent study with a one-year follow-up found a deterioration of mental health in 15% of participants from before the outbreak of the pandemic to one year later, and those who reported dysfunctional regulation strategies, such as self-harm, binge-drinking, or binge-eating, were more likely to show a worsened mental health condition [75]. Importantly, our analysis did not find any evidence of a direct link between an infection with COVID-19 (self or family members) and a deterioration of mental health. Future research should therefore rather focus on the personal and individual experiences of the pandemic-related restrictions and the associations with mental health. However, our results suggest that functional emotion regulation, high levels of self-efficacy and self-esteem, and a higher quality of life make adolescents more resilient, enabling them to deal with negative experiences such as pandemic restrictions in a healthier way. To promote resilience in school, a positive student-teacher relationship is essential, as it is associated with fewer mental health problems and also seems to moderate the relationship between cyberbullying and mental health, as well as difficulties in online learning and academic engagement, according to a recent study in China [76]. The latter study also revealed that parents and youth wish for more interactions with teachers and more support from teachers and school psychologists to address the social and emotional needs of students during COVID-19 [76].

There is a clear implication of this paper: Perhaps we need to think of schools more as a company whose staff needs support. A different philosophy and a more modern and innovative culture within the school system could help to promote individual strengths and resources. In particular, contact with and support from parents/guardians and teachers in terms of strengthening protective factors seems to be very important, as well as addressing other issues such as victimization. At the same time, we need to come up with new ways of implementing prevention programs in school. Innovation and more modern, interesting, and appealing transition methods may be key not only to attracting adolescents’ interest, but also to changing the image of prevention. Prevention programs do not just need to be conducted; they need be rethought and implemented with sustainability in mind. So far, there is no research exploring the associations not only with effectiveness, but also with longevity and change.

## Figures and Tables

**Figure 1 ijerph-20-02493-f001:**
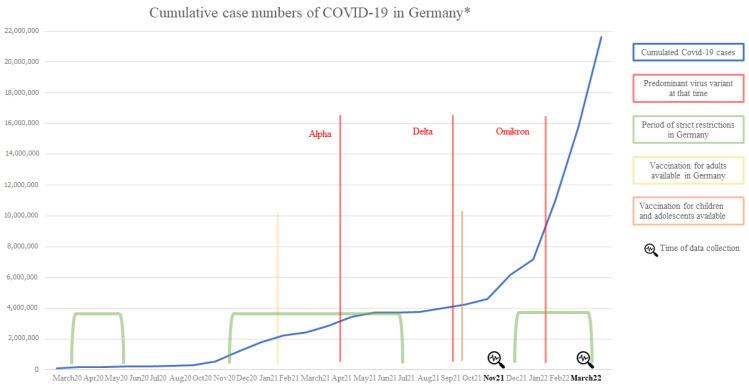
Chronology of COVID-19 in Germany (data based on WHO Health Emergency Dashboard 2022, * Data based on research of the RKI [4]).

**Figure 2 ijerph-20-02493-f002:**
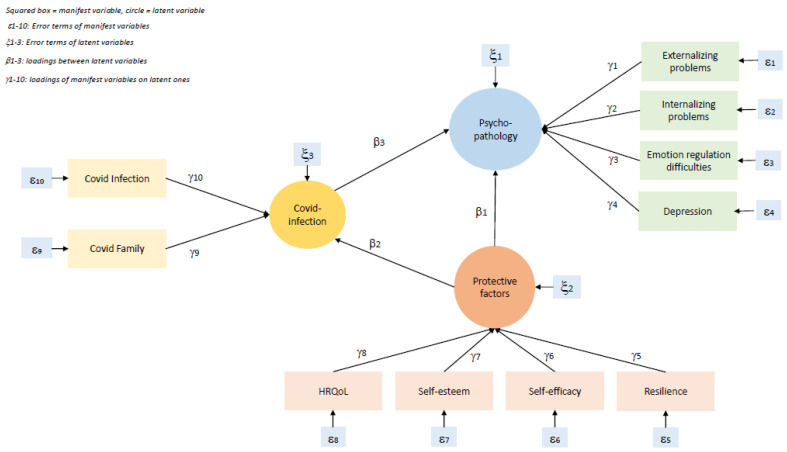
Hypothesized SEM with error terms.

**Figure 3 ijerph-20-02493-f003:**
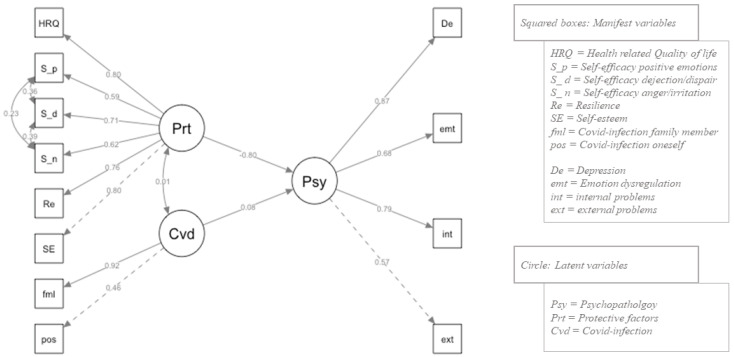
Final SEM path model.

**Table 1 ijerph-20-02493-t001:** Challenging restrictions and stress factors during the pandemic [25].

Problem	Example
Separation, loss and grief	Separation from or loss of attachment figures due to COVID-19 illnesses, no normal way of grieving possible (e.g., ban on funerals).
Social determinants of health	Financial, housing, or food problems caused by the pandemic.
Social isolation, quarantine and loneliness	School closures, loss of contact with teachers, friends, peers, disrupted contact with potential support systems, domestic conflicts and violence, fewer recreational opportunities.
Physical, intellectual, and/or learning disability	Problems accessing appropriate therapies, closure of appropriate facilities.
Disrupted daily and school routines	Higher media use, less physical activity, change in sleep patterns.
Previous traumas	Pre-existing trauma caused by, e.g., caregivers (abuse, neglect).
Previous mental health	Pre-existing psychological impairments and problems in accessing appropriate forms of therapy.

**Table 2 ijerph-20-02493-t002:** Means, standard deviations, and range for the whole sample.

		M	SD	Range
HRQoL (KIDSCREEN-10)		44.68	6.21	0–55
Self-efficacy (RESE-R)	positive emotions (S_p)	4.31	0.82	0–5
	dejection/despair (S_d)	3.65	0.97	0–5
	anger/irritation (S_a)	3.56	1.02	0–5
Self-esteem (SISE)		2.57	0.87	0–4
Resilience (CD-RISC-10)		24.70	6.95	0–40
Externalizing problems (SDQ)		5.93	3.17	0–18
Internalizing problems (SDQ)		5.24	3.28	0–19
Depression score (PHQ-9)		4.75	4.49	0–27
Emotion dysregulation (DERS-SF)		39.30	10.67	0–88
		**Yes**		
COVID-19	infection (pos)	18.98% (n = 404)		
	inf. family (fml)	46.83% (n = 997)		

**Table 3 ijerph-20-02493-t003:** Intercorrelation matrix of the manifest variables included in the SEM.

		QoL	Self-Efficacy			SE	Re	Ex	In	De	Emt	COVID-19	
			POS	DES	ANG							pos	fml
HRQoL		1											
Self-efficacy	positive emotions	0.45 **	1										
	dejection/dispair	0.50 **	0.63 **	1									
	anger/irritation	0.46 **	0.51 **	0.66 **	1								
Self-esteem (SE)		0.63 **	0.61 **	0.63 **	0.51 **	1							
Resilience (Re)		0.61 **	0.42 **	0.62 **	0.51 **	0.58 **	1						
External difficulties (ext)		−0.42 **	−0.14 **	−0.23 **	−0.40 **	−0.29 **	−0.35 **	1					
Internal difficulties (int)		−0.60 **	−0.33 **	−0.44 **	−0.37 **	−0.50 **	−0.53 **	0.41 **	1				
Depression score (De)		−0.67 **	−0.33 **	−0.43 **	−0.39 **	−0.54 **	−0.51 **	0.50 **	0.67 **	1			
Emotion dysregulation (Emt)		−0.42 **	−0.14 **	−0.33 **	−0.37 **	−0.34 **	−0.39 **	0.43 **	0.55 **	0.60 **	1		
COVID-19	infection (pos)	−0.01	0.05 *	0.05 *	0.01	0.03	0.00	0.06 *	−0.02	0.03	0.02	1	
	inf. family (fml)	−0.04	0.07 **	0.04 *	0.03	0.03	−0.01	0.08 **	0.03	0.07 **	0.04	0.43 **	1

Note: * indicates *p* < 0.05. ** indicates *p* < 0.01.

## Data Availability

Not applicable.

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
