# Peer review of "Psychopathology, Protective Factors, and COVID-19 among Adolescents: A Structural Equation Model"

_ijerph, 2023, doi:10.3390/ijerph20032493_

Round 1
Reviewer 1 Report
Dear Sir/Mam
Congrats on your work. I have attatched my comments

Reviewer 2 Report
Psychopathology, Protective factors and Covid-19 among 2 adolescents: a Structural Equation Model
This manuscript examines the relationship between psychopathology, protective factors, and covid-19 during a strict lockdown period in Germany between November 2021 and March 2022. This paper has several strengths including a large sample size and use of SEM. However, as it stands, I think there are a number of difficulties with the manuscript. First and foremost, English proofreading is absolutely necessary as grammar and phrasing made it difficult to comprehend and follow (see examples below). The discussion needs major restructuring.
My main concern is that the manuscript has some conceptual flaws as well as information lacking from the methods and results. My main take-away from this paper is that protective factors are associated with reduced psychopathology, an unsurprising finding which is well established in the literature. Those results could use more unpacking. What is it about these protective factors during the pandemic that conferred additional protection to adolescents? Was it that they were high income or were in schools with more resources and support available to students? Since the authors did not describe mean scores for each measure it is difficult to assess the demographics and personal characteristics of the sample (i.e. whether they were high in self-esteem or resilience to begin with?).
In terms of other findings, Covid-19 factors had no relationship with protective factors and a tiny non-significant association with psychopathology. However, the manifest variables making up the Covid-19 factor included infection and death only. Factors including reduced physical activity, limited social experiences, parental stress, family conflict/domestic violence, all which are mentioned in the introduction, were excluded. Whilst a Covid-19 factor adds a novel component to this model, in this case it was lacking sustenance and arguably irrelevant to an adolescent sample (i.e. adolescents are less likely to be affected by an infection). Lastly, the authors should have specified in their measures the timeframe for reporting: Were participants asked for retrospective reports of the past week, the past month? For the impact of COVID-19 where participants asked if they tested positive in the past week, month? The impact of Covid-19 might highly depend on the recency of exposure to covid and thus impact results. The timeframe when data was collected (Figure 1) suggests many changes occurred during this time including the omicron wave and tighter restrictions. The authors do not discuss this in the limitations.
Lastly, this reviewer has several conceptual issues with the way bullying victimization is handled in this analysis. First, it needs to be clear that you only assessed for victimization and NOT perpetration (bullying itself is very vague or can mean perpetration only). My main concern is that peer-victimization is a major risk factor for psychopathology (e.g. internalizing symptoms) but it is not a measure of psychopathology itself so this it is an odd decision to include it as a manifest variable. Second, were adolescents given a definition? If adolescents were asked if they were “ever bullied” this might be a very unreliable retrospective measure, particularly as data was collected during a period where bullying victimization most likely decreased. This is not discussed in the limitations.
Other major issues:
There is no mention of missing data. How did the authors deal with missing data? Did the authors check if it was missing at random before using MLM? Did they only use completed cases?
Line 227-229 of the statistical analysis– This needs to be spelled out more: “In summary, two different SEM models were calculated with the second one containing more information in the model construction, thereby being more adjusted.” Do you mean you calculated two different measurement models, one with more constraints than the other? Then those constraints need to be specified.
Authors typically start results sections with a table with demographic and mean scores for all of their measurements. Please include.
The discussion needs some serious proofreading and restructuring. It is difficult to comprehend and any recommendations are vague and could be more specific.
Line 330-331: the authors state “this impairment was and still is a debate in research and politics, as it is not clear to what extent effectiveness and efficiency are given.” This is not necessarily true. We know restrictions were imperative to curve the spread of the virus, but this was not as clear for other outcomes.
Lines 334-335: consider rewording this is difficult to comprehend: The problems in emotion regulation and the psychological stress factors associated with the limitations could be an explanation for our result.” What limitations or do you mean covid restrictions?
Line 338 – The authors talk about the “increase scores in the psychopathology of the sample” but provide no mean estimates. Was psychopathology high in your sample? Furthermore, the authors are providing explanations for increase psychopathology in their sample but they did not assess for risk factors. They should focus on WHY protective factors during the pandemic are associated with decreased psychopathology.
Line 374-477 – I don’t see how this relates: “Looking at the loadings of the manifest variables on psychopathology, there are medium to high correlations between externalizing, internalizing, depression and emotion dysregulation and psychopathology. This is in line with recent research pointing towards an increasing number of children and adolescents suffering from mental symptoms since the beginning of the Covid-19 pandemic”
Line 382 -This needs to go in a table. Overall, 45% of this sample answered yes to the question about direct bullying and 5% reported severe bullying (defined by 2 SDs above mean). And again, reported being victimized or bullied.
Line 383 – “It should be mentioned that we surveyed at German high schools, where generally a higher level of functioning can be expected from the students compared to less achievement-oriented schools” - Is this meant to say that rates of bullying victimization are higher in other schools? That is very alarming indeed.
Other minor issues:
Whilst this may be personal preference, “impairments” was not the most globally acknowledged term for covid-19 restrictions or consequences and this reviewer found it a bit confusing. Perhaps consider rewording.
Figure 1 (with German restrictions during the pandemic and data collection points) is missing a label
Abstract
Line 12- remove “of outbreak of”).
Line 17 and 18 – “…a decrease of their mental health” is grammatically incorrect, but also what you are describing is that children managed to cope well without their mental health deteriorating (i.e. MH symptoms increasing not decreasing).
Line 24 – reword “vanishingly small” or give statistics.
Introduction
Line 37 – consider revising, what do the authors mean by “previously unknown political restrictions.” Public health measures such as social distancing and lockdown are discussed after?
Line 51 – again consider revising “increase in psychopathology with associated lower mental health”. Deteriorating or poor mental health would make more sense here.
Line 61 – internalizing not “internalization problems”
Line 128-129 – revise: “…protective and risk factors concerning mental health in adolescence is increasingly raised, particularly during the pandemic of Covid-19”
Line 135 - revise: “Thus, each protective factor should be able to be attributed with some kind of mitigation of risk”
Line 140 – “including” would be more clear than “involving”
Line 145- Are the authors looking at peer-victimization or bullying perpetration? Please be clear as your measure seems to be only peer-victimization. Furthermore, in line 242 it appears bullying victimization is referred to as “mobbing”. Please be consistent with your terminology.
Measures
Line 151 – Specify dates (e.g. November 2020 -March 2021). Was data continuously collected between these time points? Please make that clear.
All measures need to specify the timeframe for reporting. Were they asking for retrospective reports of the past week, the past month?
Analysis
Line 253-257 – This is confusing, please reword: “A model has an acceptable fit with a value greater than 0.9 and a good model fit, if the TLI assumes a value greater than 0.9 and the CFI a value greater than 0.95”. Similar in 256-258: “For the RMSEA, different threshold literature, a common threshold value for the RMSEA is 0.06, which indicates a good model 257 fit to the data.”
Results
Line 293 – Correlations are moderate to strong.
Line 307 – should specify if this main correlation is significant.
Discussion
Other limitations mentioned elsewhere need to be included.
Conclusion
Line 417 – consider revising “protective factors can have a negative impact on psychopathology” (e.g. increase in protective factors is associated with a decrease in psychopathology symptoms)
Reviewer 3 Report
Although it is an important research topic, it needs improvement in the following points.
1. Although the reason for using SEM is to statistically confirm the role of mediating variable, there was no clear presentation of the mediating variable of this study.
2. Looking at Figure 2, it seems that covid 19 was probably set as a mediating variable, but there was no discussion of the results of any previous studies on the effect of protective factors on covid 19 in the theoretical background.
3. Moreover, it is difficult to understand that only correlation values are presented as a result of SEM. It is necessary to present the results of the regression analysis on the relationship between the variables set by the researcher.
4. Discussion section needs to contain lots of rich arguments, which is very worthwhile. The points would read better with a clear that what are the summaries of the main findings, what research implications and suggestions for future research are discussed, and what kinds of program/policy implications can be made.
Round 2
Reviewer 2 Report
The authors have done substantial improvements to the manuscript, namely English proofreading, that allowed this reviewer to comprehend the manuscript. Despite considering the feedback and improvements in writing style and restructuring, this reviewer does not think the manuscript is ready for publication and could use further proofreading. The authors should have conceptualized this study better, provided more accurate interpretation of their results and recommendations, and carefully revised their manuscript for errors. See specific points below:
My main concern in the methods is how you created your latent variables: a) psychopathology including victimization; which is not theoretically or empirically correct and b) the covid variable; the authors are giving the same weight to being infected with covid and a family member dying in the same variable. To me this doesn’t sound right. The correlations are tiny and the variables don’t even correlate moderately with each other. This construct is probably not very good because of the death variable and probably should have been excluded from the analysis. Typically, if you have only three variables you should have high loadings > 04. I also don’t see why the authors “hypothesized a negative influence of COVID-19 on psychopathology and a positive influence of protective factors on both psychopathology and on the impact of COVID-19.” Seeing as 2/3 items in the covid variable refer to infection, I am not sure why you are examining an association between protective factors such as self-esteem or resilience and getting infected with the virus? I don't think there is much conceptual justificatin for this. If anything, you could have looked at how protective factors mediate the relationship between covid and psychopathology. Lastly, the authors “collected data at two time points (fall 2021 and spring 2022)”, in which case they should have looked at these two time points separately as the context might have been significantly different with regards to the pandemic.
Regarding interpretation, I think the author’s hypothesis are not in line with their interpretations and recommendations (see examples below), and in some cases the model fit indices are described as "good", when they are not based on the standards they present.
Abstract:
Lines 26-27. Without pre-pandemic data, I wouldn’t say covid 19 had an impact on psychopathology.
Lines 27 What are external pandemic-related factors? Do you mean other factors not captured in this study
Line 30 – you do not look at risk groups but this makes me think you are.
Overall the conclusion can be misleading as baseline infections and deaths/bereavement was low in your sample so claiming that the virus doesn’t have an impact on young people’s mental health might be misleading. I think you should start by saying something along the lines of “Whilst infection rates and deaths were low in our sample, XXX.”
Figure 1. Perhaps include tick marks on the x axis
Introduction
48-49 The reference here doesn’t necessarily follow what you are saying – consider deleting “as they show a higher risk of developing mental health problems (ref).” The next reference (meta-analysis) is good and covers your argument.
Lines 68-73: There is a lot of confusion here. You start by saying that compliance with stay at home orders (in other words, restriction) is a protective factor for mental health, then that adolescents are more concerned about the restrictions which are associated with a deterioration of mental health. Then that adolescents who were infected by covid were more likely to be mentally healthy? Not sure what message you are trying to get across.
Line 89 – developmental stages not steps. Consider deleting the sentence in lines 88-90. Not sure it contributes anything to you aims.
Lines 108 – stress factors did not affect all children and adolescents equally with a massive amount of research showing how disadvantaged young people fared worse.
Lines 129 – peer-victimization STILL considered psychopathology rather than a risk factor? This reviewer thinks this is conceptually a BIG problem (already stated in previous review).
Lines 131 –authors are not looking at vulnerable groups only a group of children and adolescents? Even in your sample they seem to be doing okay at baseline. Suggest you remove.
Methods
Line 25 – what risk factors are the authors referring to? They do not measure any according to the methods. Then also revise Line 12, and any other reference to risk factors.
Measures of model fit Line 243-244: “Model fit is deemed to be acceptable if the TLI is greater than 0.9 and good if the TLI is greater than 0.9 and the CFI greater than 0.95.” Revise parameter (second 0.9).
Hypothesis and aims Lines 227 -228: “We hypothesized a negative influence of COVID-19 on psychopathology and a positive influence of protective factors on both psychopathology and on the impact of COVID-19.” Seeing as 2/3 items in the covid variable refer to infection, I am not sure why you are examining an association between protective factors such as self-esteem and quality of life or resilience and getting infected with the virus. If anything you could have looked at how protective factors mediate the relationship between covid and psychopathology. Conceptually I am not sure this makes sense.
Figure 2 – what is the NO column number representing in the covid-19 variable? Not clear
Final model Line 276 – the model does not have a good fit. It has an acceptable fit. CFI and TLI is not even .90. Model 2 does not have a great fit either.
Discussion
I think the interpretation are too strong and misleading, given the number of limitations and statistical issues in the methods.
Line 319 – that sounds like a mediation analysis with covid being the mediator.
Line 317 – you don’t use SEM for this, this is just a correlation analysis
Line 320- this is repetitive (needs proofreading) and has no references: “This finding is well established in the literature, but at the same time it has very clear implications for prevention in general, and specifically for universal prevention. Targeting protective factors and promoting resilience indeed seem to help, even at a young age, or even more crucially in young people.”
Lines 325-327: “This finding is in line with recent research suggesting that adolescents are more concerned about government-mandated restrictions than about the virus itself and are more likely to remain mentally healthy when they are infected with the virus.” You found a POSITIVE correlation between covid and psychopathology, correct? Meaning more difficulties with covid (e.g. getting infected) was associated with higher levels of psychopathology. How is that is line with the study results?
Lines 328-332, and then again in 354-355: “Moreover, the correlation between the protective factors and the impact of COVID-19 was zero, suggesting no mutual influence of these two latent variables. Again, this confirms our hypothesis that the observed deterioration in mental health among adolescents is attributable not to the virus itself but rather to the stress factors summarized and suggested by Rider et al.” That is not what you hypothesized? You said (line 227): “We hypothesized a negative influence of COVID-19 on psychopathology and a positive influence of protective factors on both psychopathology and on the impact of COVID-19.” The authors do not hypothesize that deterioration in mental health are not directly related to covid (line 354).
Line 333: “In sum, the restrictions due to the virus, and not the virus itself or virus-related fears, seem to be responsible for the deterioration in adolescents’ mental health over the last two years.” You don’t look at covid restrictions so not sure how this is the conclusion.
Lines 370-399: This reviewer previously mentioned that some results regarding protective factors needed more unpacking, specifically what was it about protective factors that conferred additional protection to adolescents in this study. I think the authors misunderstood this comment to mean they needed to include more background on the effect of protective factors on mental health. That is well known already. This reviewer was suggesting the authors unpack their OWN study results – why did self-esteem and resilience or self-efficacy confer protection to your sample’s psychopathology. Is it that they were really low on mental health problems to begin with? Did you sample come from affluent backgrounds? (what you describe in line 405 would be more relevant to this paragraph). Identifying what worked in your sample and making recommendations based on what you found in your sample is more valid than giving a literature review on protective factors you did not study. In this paragraph there is no link between your study results and other study results.
409- victims of victimization is not typically used. Consider rephrasing to children who are victimized.
Line 415-417: The recommendations made are vague, for example: “This issue needs to be addressed and targeted to a greater extent, and teachers and students should be supported accordingly. Such an endeavour does not appear to be easy and may require a different type of intervention or prevention to those implemented so far.”
